

# Chemical profile and pancreatic lipase inhibitory activity of *Sinobambusa tootsik* (Sieb.) Makino leaves

Xiao-Lin Qiu[1] and Qing-Feng Zhang[1,2]

[1] College of New Energy and Environmental Engineering, Nanchang Institute of Technology, Nanchang, China
[2] Jiangxi Key Laboratory of Natural Product and Functional Food, College of Food Science and Engineering, Jiangxi Agricultural University, Nanchang, China

## ABSTRACT

**Background**. *Sinobambusa tootsik* (Sieb.) Makino (*S. tootsik*) is one species of bamboo distributed in China, Japan and Vietnam. The chemical profile of its leaves and its potential application was unknown yet.

**Methods**. The chemical profile of *S. tootsik* was studied by HPLC and UPLC-DAD-QTOF-MS. The *S. tootsik* extract was prepared by extraction with 50% aqueous ethanol, followed by H103 macroporous resins adsorption and desorption processes. Pancreatic lipase inhibitory activity was determined using *p*-nitrophenyl palmitate as the substance, which was hydrolyzed by lipase to form coloured *p*-nitrophenol.

**Results**. Eighteen compounds were identified in *S. tootsik*. Most of them were the *C*-glycosylated derivatives of luteolin and apigenin, such as isoorientin, isoorientin-2″-*O*-rhamnoside and isovitexin. Isoorientin-2″-*O*-rhamnoside was the most dominant flavonoid in the sample. *S. tootsik* extract was prepared through resin adsorption/desorption with yield of $1.12 \pm 015\%$ and total flavonoids content of $82 \pm 2$ mg/g (in term of isoorientin). The extract exhibited pancreatic lipase inhibitory activity with IC50 value of 0.93 mg/mL.

**Conclusion**. The chemical profile of *S. tootsik* leaves was uncovered for the first time. *C*-glycosyl flavonoids were the main constituents in the plant. The extract exhibited pancreatic lipase inhibitory activity and may have potential for use as a food supplement for controlling obesity.

Corresponding author
Qing-Feng Zhang, zhqf619@126.com

## INTRODUCTION

Bamboo is a valuable plant distributed all over the world with more than 1,500 species. The bamboo shoots of some species, e.g., *Phyllostachys heterocycla* cv. pubescens (*P. heterocycla*), are eaten as vegetable, while the leaves are used as herbal material in China. The flavonoids extract of some bamboo species (*Phyllostachys* genus) were approved as a food antioxidant and food resource in China (*Wang et al., 2012*). The pharmacological activities of bamboo leaves arise from the presence of phytochemicals. For instance, five *C*-Glycosyl flavones were isolated from *Fargesia robusta* (*Van Hoyweghen et al., 2010*). Three chlorogenic acid derivatives were isolated from *Phyllostachys edulis* and the antioxidant activity was

evaluated (*Kweon, Hwang & Sung, 2001*). *Wang et al. (2012)* isolated three isoorientin derivatives from *Bambusa. textilis* McClure. Previous, we identified twelve compounds in the leaves of *Bambusa multiplex* cv. Fernleaf (*B. multiplex*), and found that *C*-glycosyl flavonoids including vitexin, isovitexin, isoorientin and its derivatives, are the main chemical constitutes of the plant (*Qiu & Zhang, 2019*). *Sinobambusa tootsik* (Sieb.) Makino (*S. tootsik*) is one species of bamboo distributed in China, Japan and Vietnam. To the best of our knowledge, the chemical profile of its leaves has not been studied yet. To further uncover its potential application, the chemical composition of *S. tootsik* was studied by HPLC and UPLC-QTOF-MS in the present study. Furthermore, the pancreatic lipase inhibition activity of its extract was studied.

## MATERIALS AND METHODS

### Chemicals and plant materials

Leaves of *S. tootsik* were collected in Jiangxi Agricultural University (with east longitude of $115°50'$ and northern latitude of $28°46'$) on Mar. 2019. The plant material was authenticated based on morphological characters by Prof. Qing-Pei Yang (Jiangxi Agricultural University), and the voucher specimen was deposited in Jiangxi Key Laboratory of Natural Product and Functional Food. The leaves was dried at 60 °C and smashed to filter through 40 mesh sieve. Isoorientin standard (>98%) was purchased from Beijing Solarbio Science & Technology Co., Ltd (Beijing, China). HPLC grade acetonitrile was purchased from Anhui Tedia High Purity Solvents Co., Ltd. (Anqin, China). Porcine pancreatic lipase (extract powder, 15–35 units/mg, catalog number L111237) was purchased from Aladdin Chemistry Co. Ltd. (Shanghai, China; http://www.aladdin-e.com). Milli-Q water was used throughout the study. All other reagents used were analytical grade.

### Sample extraction

A 0.1 g aliquot of *S. tootsik* powder was mixed with 5.0 mL of 50% aqueous ethanol. After sonicating for 30 min in a bath sonicator (100 W, 45 kHz, Kunshan, China), the mixture was centrifuged at 1,100 *g* for 5 min. The supernatant was filtered by 0.22 mm pore size filter and then used for HPLC and UPLC-DAD-QTOF-MS analysis.

For *S. tootsik* extract preparation, 50 g of *S. tootsik* sample was extracted for twice with 500 mL of 50% aqueous ethanol each time. After centrifugation at 1,100 g for 5 min, the supernatant was combined together. The extract was condensed to about 500 mL by vacuum rotavapor at 50 °C. The concentrates was two times diluted by water. Then, the extract was pumped through a fixed bed of H103 macroporous resin with diameter of 1.5 cm and height of 40 cm in a glass column. The flow rate was 10 mL/min. After adsorption, the fixed bed was desorbed with 4 BV of 90% ethanol with flow rate of 5 mL/min. The eluent was concentrated by vacuum rotavapor at 50 °C and then lyophilized to obtain the extract.

### UPLC-DAD-QTOF-MS analysis

The chemical identification was performed on a QTOF 5600-plus mass spectrometer equipped with Turbo V sources and a Turbolonspray interface (AB Sciex Corporation, Foster City, CA, USA) coupled to a Shimadzu LC-30A UPLC-DAD system (Shimadzu

Corporation, Kyoto, Japan). Acquity UPLC BEH C18 column (2.1 mm × 100 mm, 1.7 μm, Waters) was used. The flow rate was 0.3 mL/min with injection volume of 1 μL and column temperature of 40 °C. The mobile phase was acetonitrile (A) and 0.1% formic acid aqueous solution (B) using a linear gradient program of 0–30 min, 5–40% (A). The mass spectrometer was operated in the negative ion mode. Ultrapure nitrogen was used as the ion source gas 1 (50 psi), ion source gas 2 (50 psi), and curtain gas (40 psi). The Turbo Ion Spray voltage and temperature were set at −4,500 V and 500 °C, respectively. Declustering potential, collision energy, and collision energy spread were set at 100 V, −40 V, and 10 V, respectively. Data acquisition was performed with Analyst 1.6 software (AB Sciex).

## HPLC quantification analysis

The HPLC Analysis was performed on an Agilent 1260 HPLC system equipped with an autosampler and DAD detector. A Symmetry C18 column (250 mm × 4.6 mm i.d., 5 μm; Waters, USA) was used as the stationary phase. The mobile phase consisted of acetonitrile (A) and 0.1% acetic acid aqueous solution (B). The flow rate was 1 mL/min with linear gradient program of 0–30 min, 1–40% A; 30–35 min, 40% A. Detected wavelength was 349 nm with injection volume of 10 μL and column temperature of 40 °C.

## Pancreatic lipase inhibitory activity assay

Pancreatic lipase inhibitory activity was determined using $p$-nitrophenyl palmitate ($p$-NPP) as the substance, which was hydrolyzed by lipase to form $p$- nitrophenol with maximum absorption around 405 nm. Lipase (10 mg) was dissolved in 5 mL Tris-buffer (50 mM, pH 8, containing 0.1% gum arabic powder and 0.2% sodium deoxycholate). The mixture was stirred for 15 min and centrifuged at 1,800 $g$ for 10 min. The clear supernatant was used for the assay. Briefly, in a 96-well microplate, 30 μL Tris-buffer, 150 μL enzyme and 10 μL *S. tootsik* extract (dissolved in 50% ethanol) were mixed together. The mixture was incubated at 37 °C in the microplate reader for 20 min. Then, 10 μL of 10 mM $p$-NPP pre-incubated at 37 °C was added to start the reaction. The absorbance was determined under 405 nm for 20 min with interval of 1 min. The absorbance growth slope (V) which represented the enzyme activity was calculated by linear regression.

$$\text{Lipase inhibition activity (\%)} = \frac{V_b - V_s}{V_b} \times 100.$$

Where $V_b$ and $V_s$ were the enzyme activity in the absence and presence of *S. tootsik* extract, respectively. Orlistat was used as the positive control. The same reaction mixture but without lipase was used as the negative control, in which no absorbance change was found.

## Fluorescence quench measurements

A 1.0 mL aliquot of the lipase solution was mixed with 4 mL of Tris-buffer. Subsequently, 0, 5, 10, 15, 20, 25 and 30 μL of *S. tootsik* extract (10 mg/mL in 50% aqueous ethanol) was added, respectively. The fluorescence spectra of the mixture was recorded between 300 to 450 nm under the excitation wavelength of 280 nm. A 970 CRT spectrofluophotometer (Shanghai Scientific Instruments Limited Company, Shanghai, China) was used, and the excitation and emission bandwidths were set at 10 nm.

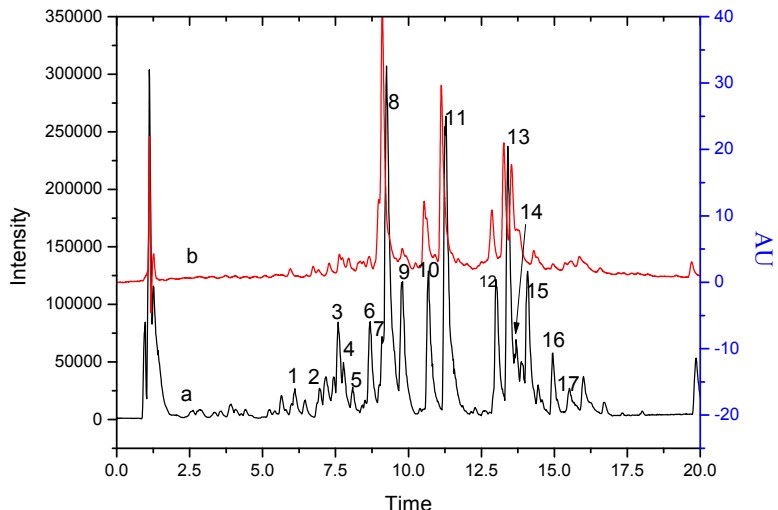

**Figure 1** **The chromatograms of *S. tootsik* detected by QTOF-MS (A) and DAD (B) after UPLC separation.** Line a: base peak chromatogram of QTOF-MS; Line b: detected at 349 nm.

## Statistical analysis

Data were expressed as the mean ± standard deviation (SD) of triplicates. Statistical analysis, plotting, and curve fitting were performed by Origin 7.0 (Origin Lab Co., Northampton, MA, USA).

## RESULT AND DISCUSSION

### Chemical profile of *S. tootsik*

Figure 1 corresponds to the chromatograms of *S. tootsik* detected by QTOF-MS and DAD after UPLC separation. By the QTOF-MS detector, the molecular mass of each peak and its further MS$^2$ spectrum was obtained. The chemical identification was accomplished by comparing these information with published literature. The details were listed in Table 1. A total of 18 components were identified. Most of them were the *C*-glycosylated derivatives of luteolin and apigenin, such as isoorientin, isoorientin-2″-*O*-rhamnoside and isovitexin. Besides, some other *C*- glycosyl and *O*-glycosyl flavonoids were found, such as isoscoparin-*O*-deoxyhexoside and kaempferol-*O*-glucoside. Two non-flavonoid compounds, feruloylquinic acid and roseoside, were also found. *S. tootsik* belongs to the family of *Poaceae*. Many studies showed that the main secondary metabolite found in the leaves of *Poaceae* plants were *C*-glycosyl flavonoids, for instance, barley, maize, wheat, rice, etc. (*Brazier-Hicks et al., 2009*; *Ferreres et al., 2008*). Previously, we had studied the chemical constituents in the leaves of *B. multiplex*, another bamboo species (*Qiu & Zhang, 2019*). It was found that *C*-glycosylated derivatives of luteolin and apigenin were the main components of both species. However, the specific flavonoids between the two plants were different. Only apigenin 6-*C*-pentoside-8-*C*-glucoside, isoorientin and isovitexin were found in the both species.

**Table 1** Mass characterizations of main peak in the chromatogram of *Sinobambusa tootsik* (Sieb.) Makino by UPLC-QTOF-MS.

| Peak no. | RT (min) | [M–H]⁻ (m/z) | Fragment ions (m/z) (% base peak) | Proposed structure | Reference |
|---|---|---|---|---|---|
| 1 | 6.10 | 367.1033 | 193(45), 134(100), 117(10) | Feruloylquinic acid | *Qiu, Guo & Zhang (2018)* |
| 2 | 6.97 | 609.1449 | 519(25), 489(60), 399(70), 369(100) | Quercetin-3-*O*-robinobioside | *Iswaldi et al. (2011)* |
| 3 | 7.60 | 431.1915 | 385(15), 205(35), 153(100) | Roseoside | *Spínola, Pinto & Castilho (2015)* |
| 4 | 7.78 | 519.1707 | 325(10), 265(33), 223(60), 205(100), 190(55) | Unidentified | |
| 5 | 8.09 | 489.159 | 223(40), 205(100), 190(80), 164(27) | Unidentified | |
| 6 | 8.69 | 371.098 | 121(100), 249(45), 231(10) | Unidentified | |
| 7 | 9.14 | 563.1393 | 353(100), 383(65), 443(45), 473(32) | Apigenin 6-*C*-pentoside-8-*C*-glucoside | *Ozarowski et al. (2018)* |
| | 9.14 | 447.0924 | 327(100), 357(70), 297(55), 285(35) | Isoorientin (luteolin 6-*C*-glucoside) | *Figueirinha et al. (2008)* |
| 8 | 9.26 | 593.1504 | 298(100), 473(85), 327(55), 309(40), 357(35), 429(25) | Isoorientin-2″-*O*-rhamnoside | *Ibrahim et al. (2015)* |
| 9 | 9.78 | 613.213 | 181(100), 387(85), 166(30), 205(25), 399(20) | Unidentified | |
| 10 | 10.68 | 533.128 | 353(100), 383(90), 443(50), 473(40), 365(25), 297(23) | Apigenin 6, 8-di-*C*-pentoside | *Ozarowski et al. (2018)* |
| | 10.68 | 577.1546 | 293(100), 413(35), 323(15), 311(15), 457(10), | Isovitexin-2″-*O*-rhamnoside | *Ibrahim et al. (2015)* |
| | 10.71 | 431.0986 | 311(100), 341(35), 283(75) | Isovitexin (apigenin 6-*C*-glucoside) | *Ibrahim et al. (2015)* |
| 11 | 11.27 | 607.1649 | 323(100), 443(40), 308(20), 341(15) | Isoscoparin-*O*-deoxyhexoside | *Ozarowski et al. (2018)* |
| | 11.29 | 447.091 | 285(100)) | Kaempferol-*O*-glucoside | *Singh et al. (2011)* |
| 12 | 12.89 | 561.1595 | 561(100), 457(30), 399(14), 337(18), 295(40) | Chrysin 6-*C*-deoxyhexoside-7-*O*-glucoside | *Ozarowski et al. (2018)* |
| | 13.01 | 637.1759 | 329(100), 314(15), 299(10) | 3, 4-Dihydroxy-5,6-dimethoxy-7-*O*-rutinoside flavone | *Han et al. (2007)* |
| 13 | 13.41 | 547.1446 | 293(100), 383(85), 341(35), 311(28) | Apigenin 6-*C*-[2″-*O*-deoxyhexoside]-pentoside | *Ozarowski et al. (2018)* |
| 14 | 13.69 | 577.1546 | 311(100), 415(50), 397(15) | Apigenin-6-*C*-deoxyhexoside-7-*O*-glucoside | *Ozarowski et al. (2018)* |
| 15 | 14.08 | 575.1392 | 325(100), 297(100), 411(100), 337(70), 285(70), 367(55) | "X"-*O*- Rhamnosyl *C*- (6-deoxy-pento-hexos-ulosyl) luteolin | *Figueirinha et al. (2008)* |
| 16 | 14.45 | 577.1549 | 311(100), 298(70), 415(70), 473(50), 327(35) | Apigenin-8-*C*-deoxyhexoside-7-*O*-glucoside | *Ozarowski et al. (2018)* |
| 17 | 15.52 | 559.1441 | 457(10), 395(95), 321(100), 309(25), 293(50), 281(30), 269(60) | Apigenin-8-*C*-[6-deoxy-2-*O*-rhamnosyl]-xylo-Hexos-3-uloside | *Bezerra et al. (2016)* |
| 18 | 16.00 | 589.1554 | 425(100), 351(65), 325(35), 299(35) | Unidentified | |

Figure 2A was the HPLC chromatogram of *S. tootsik* detected at 349 nm. With the results of UPLC-DAD-Q-TOF-MS analysis (Fig. 1), the five main peaks in the HPLC chromatogram were identified. The peak of isoorientin (peak 2) was further validated by comparing the retention time and UV spectra with standard marker. Form the peak area, it was found that isoorientin- 2″-*O*-rhamnoside (peak 1) was the most dominant flavonoid in *S. tootsik,* followed by isoscoparin-*O*-deoxyhexoside (peak 4) and apigenin 6-*C*-[ 2″-*O*-deoxyhexoside]-pentoside (peak 5). This was also different from *B. multiplex,* in which isoorientin was the most dominant flavonoid, followed by isovitexin (*Qiu & Zhang, 2019*). The UV spectra of the five peaks were presented in Fig. 2B. It was found that the UV spectra of peaks 1, 2 and 4 were very similar with maxmium absorption around 348 nm, while the maxmium absorption of peaks 3 and 5 was around 338 nm.

### *S. tootsik* extract preparation

Through 50% aqueous ethanol extraction, followed by H103 macroporous resins adsorption and desorption processes, the yield of *S. tootsik* extract was $1.12 \pm 015\%$. HPLC analysis showed that the chemical profile was unchanged (Fig. 2). However, the chemical content reflected by peak area were about 29.8 times increased. Besides isoorientin, most of the other flavonoids identified in *S. tootsik* were market unavailable. The calibration curves of isoorientin were $Y = 24.82X$, with correlation coefficient of 0.999, where Y was the peak area and $X$ was concentration of isoorientin (5–200 µg/ml). By submitted the area sum of peak 1 to 5 to the calibration curves, the total flavonoids content in *S. tootsik* extract was calculated as $82 \pm 2$ mg/g in term of isoorientin.

### Pancreatic lipase inhibitory activity of *S. tootsik* extract

Obesity is becoming one of the biggest threats to human health around the world. Before being absorbed by the small intestine, food fats need first be hydrolyzed by lipase into monoglycerol and free fatty acids. Thus, the inhibition of lipase, especially pancreatic lipase, could effectively reduce the absorption of fat in the diet, thereby controlling and treating obesity (*Bialecka-Florjanczyk et al., 2018*; *Birari & Bhutani, 2007*; *Buchholz & Melzig, 2015*; *Yun, 2010*). Thus, the finding of lipase inhibitor from natural source is getting more and more attention. Many flavonoids from plant source show pancreatic lipase inhibitory activity, such as luteolin, genistein, hyperin, kaempferol, etc. (*Buchholz & Melzig, 2015*). *Lee et al. (2010)* found that the *C*-glycosylated derivatives of luteolin on A-ring exhibited much stronger pancreatic lipase inhibitory activity than luteolin. The main identified flavonoids in *S. tootsik* were the *C*-glycosylated derivatives of luteolin and apigenin. Thus, the pancreatic lipase inhibitory activity of *S. tootsik* extract was studied in the present study. The result showed that the pancreatic lipase inhibitory activity of *S. tootsik* extract steadily increased with the concentration, and the IC50 value was about 0.93 mg/mL (Fig. 3). In comparison, the IC50 value of Orlistat, the clinically approved pancreatic lipase inhibitor, was 74 ng/mL. Many plant extracts with components of saponins, phenolic acids, and/or flavonoids, possess pancreatic lipase inhibitory effects (*Buchholz & Melzig, 2015*; *De la Garza et al., 2011*). For instance, *Crocus cancellatus* subsp. *damascenus* extract with main constituents of catechin, ferulic and caffeic acids, induced 50.39% of inhibition of lipase

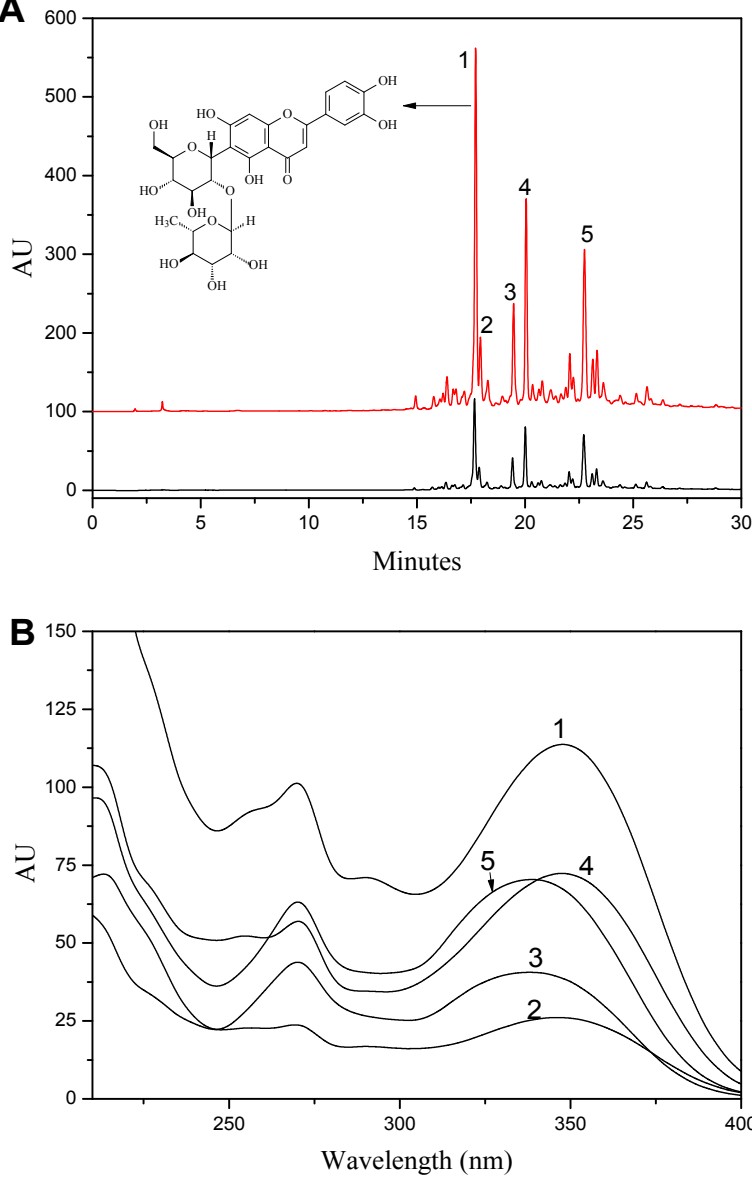

**Figure 2  A: HPLC chromatogram of *S. tootsik* extract before (A) and after (B) resin purification; B: The UV spectra of peak 1–5.** Peaks: 1, Isoorientin-2″-*O*-rhamnoside; 2, Isoorientin; *3,* Isovitexin-2″-*O*-rhamnoside; 4, Isoscoparin-*O*-deoxyhexoside; 5, Apigenin 6-*C*-[2″-*O*-deoxyhexoside]-pentoside.

activity at 5 mg/mL (*Loizzo et al., 2016*). The acetic extracts of *Aronia melanocarpa* L. and its cyanidin-3-glucoside fraction exhibited pancreatic lipase inhibitory activities with IC50 values of 83.45 and 1.74 mg/mL, respectively (*Worsztynowicz et al., 2014*). The IC50 value of *Moricandia arvensis* (L.) DC methanolic extract with main constituents of flavonoid

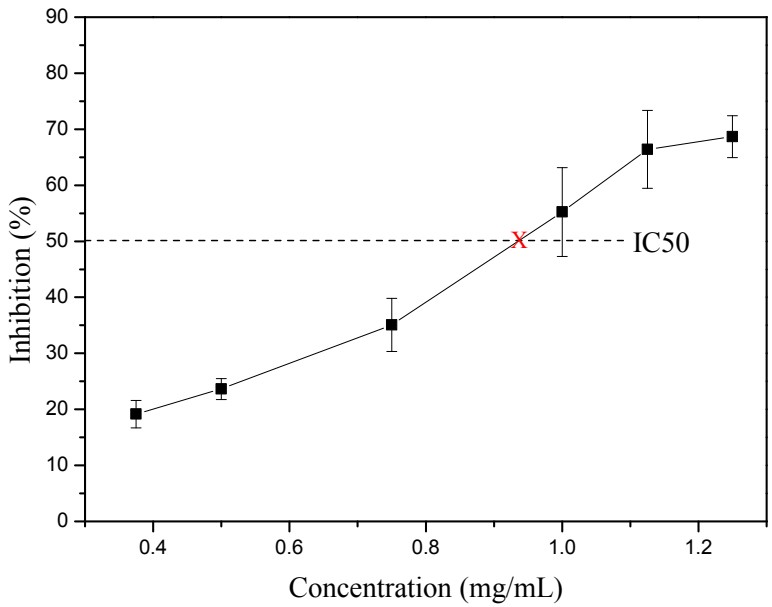

**Figure 3** **The lipase inhibitory activity of *S. tootsik* extract.**

glycosides was 2.06 mg/mL, while the IC50 value of Orlistat was 0.018 mg/mL in the study (*Marrelli et al., 2018*).

Fluorometric analysis showed that the addition of *S. tootsik* extract could gradually quench the endogenous fluorescence of pancreatic lipase (Fig. 4). Furthermore, it also caused the red shift of maximum emission wavelength. These phenomenons implied that the flavonoids in *S. tootsik* extract could bind on the enzyme. If the ligand is a monomeric compound, the fluorescence titration results can be further used to calculated the binding constant and binding site of the complexes using Stern-Volmer equations. However, in the present study, the *S. tootsik* extract is a mixture without definite molecular weight, and Stern-Volmer equations can't be applied.

Although the lipase inhibitory activity of *S. tootsik* extract was far weaker than Orlistat, as an abundant and safe natural product, it may also have potential to be used as a food supplement for obesity controlling. The in *vivo* study of its anti-obesity is in progress in our Lab.

## CONCLUSION

The chemical profile of *S. tootsik* was studied by HPLC and UPLC-DAD-QTOF-MS. Eighteen compounds were identified, most of them were the *C*-glycosylated derivatives of luteolin and apigenin, such as isoorientin, isoorientin-2″-*O*-rhamnoside and isovitexin. Isoorientin-2″-*O*- rhamnoside was the most dominant flavonoid in the sample. *S. tootsik* extract was prepared through resin adsorption/desorption with yield of 1.12 ± 0.15% and total flavonoids content of 82 ± 2 mg/g (in term of isoorientin). The extract exhibited pancreatic lipase inhibitory activity with IC50 value of about 0.93 mg/mL.

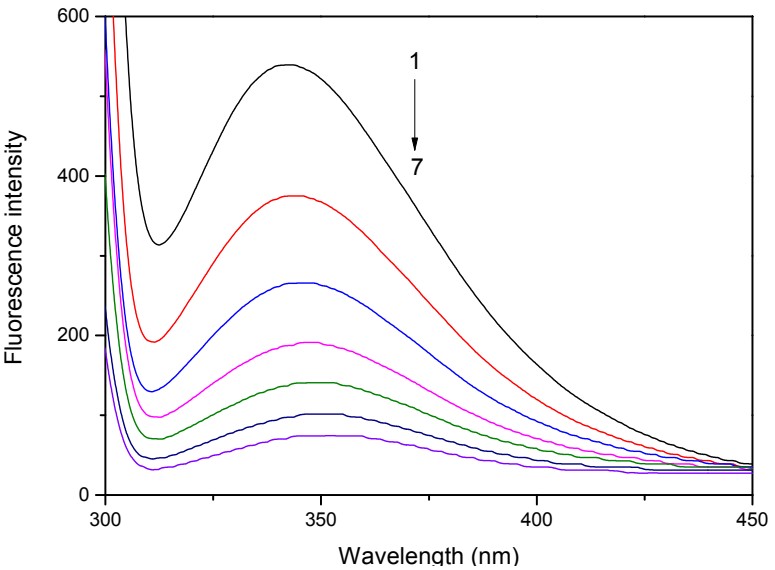

**Figure 4** **The effect of *S. tootsik* extract on fluorescence emission spectra of pancreatic lipase.** The concentrations of *S. tootsik* extract from 1 to 7 were 0, 10, 20, 30, 40, 50, 60 µg/mL, respectively.

### Funding
This work was supported by the National Natural Science Foundation of China (Grant Number 31760461). The funders had no role in study design, data collection and analysis, decision to publish, or preparation of the manuscript.

### Grant Disclosures
The following grant information was disclosed by the authors:
National Natural Science Foundation of China: 31760461.

### Competing Interests
The authors declare there are no competing interests.

### Author Contributions
- Xiao-Lin Qiu performed the experiments, analyzed the data.
- Qing-Feng Zhang conceived and designed the experiments, contributed reagents/materials/analysis tools, prepared figures and/or tables, authored or reviewed drafts of the paper, approved the final draft.

### Data Availability
The raw measurements are available in the Supplemental File.

## Supplemental Information

Supplemental information for this article can be found online at http://dx.doi.org/10.7717/peerj.7765#supplemental-information.

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
