# Peer review of "Chemical profile and pancreatic lipase inhibitory activity of Sinobambusa tootsik (Sieb.) Makino leaves"

_PeerJ, doi:10.7717/peerj.7765_

## Round 0.1 · original submission · Major Revisions

Besides the comments of the reviewers, a PDF with specific comments on the manuscript is attached. Main concerns are details on the production of the extract. Also, the analysis (calculated graph vs. experimental data) to obtain the IC50 for porcine lipase is important to show. A table or a comparison between the IC50 values for other plant extracts with lipase inhibiting activity and vs. the Orlistat control. Typos and formatting details are also mentioned.
Please provide a detailed rebuttal letter aside your amended manuscript.

Reviewer 1 ·

Basic reporting

The editors recommend two separate chapters Results and Discussion. Here they have been combined, which I think is right, because in this type of work, which is actually based on analysis and compounds identification, the discussion can not take up much place if the identification is not in doubt.

Experimental design

The work is in the journal's profile. Separation and identification of compounds have been properly planned and performed.

Validity of the findings

This type of research always expands knowledge in the field of etnomedicine. In this case, it is possible to develop new dietary supplements to reduce obesity. The tested extract has much lower effectiveness compared to orlistat, so it should be considered only in terms of a natural functional dietary supplement. That's why, in my opinion, it would be necessary to add some references to the reviews work, e.g.
Possible anti-obesity therapeutics from nature--a review. Yun JW. Phytochemistry. 2010 Oct;71(14-15):1625-41.
Synthetic and Natural Lipase Inhibitors.Bialecka-Florjanczyk E et al. Mini Rev Med Chem. (2018) 18(8):672-683.

Additional comments

The proper methods were used and the compounds were correctly identified. Measurements of lipase activity indicate the potential practical use of the extract in the treatment of obesity.
That's why I would like to supplement the references with some other reviews in this field, e.g.
Possible anti-obesity therapeutics from nature--a review. Yun JW. Phytochemistry. 2010 Oct;71(14-15):1625-41.
Synthetic and Natural Lipase Inhibitors.Bialecka-Florjanczyk E et al. Mini Rev Med Chem. (2018) 18(8):672-683.

Reviewer 2 ·

Basic reporting

- Some more updated references should be included. There are recent papers (see for example Marrelli et al., 2016 and 2018) dealing with natural phenolics having pancreatic lipase inhibitory effects.

- English should be revised

Experimental design

Experiments are generally well designed and well described.

HPLC farce of the extract seems indeed quite simple, while in Table 1, about 23 compounds are reported. I would suggest to number the identified ones and add their UV spectra in the table. UV is also very important to confirm identity of flavonoids.

Validity of the findings

Authors should include and compare their data with more recent results from other plants which show inhibition of pancreatic lipase.

Additional comments

Sistematic name of plants (genus species, but not Author or variety) should be written in italic: Authors should check for this in all the text.

---

## Round 0.2 · Minor Revisions

Please attend the corrections made in the attached PDF. These are mostly to change to small case and italicize some chemical names. Those corrections are needed to further proceed.

Reviewer 2 ·

Basic reporting

Authors have answered to previous comments and manuscript is now acceptable for publication. I have annotated some minor corrections to be done.

Experimental design

See above. Authors have improved the manuscript

Validity of the findings

See above

Additional comments

Manuscropt has been improved and is now acceptable for publication.

Annotated reviews are not available for download in order to protect the identity of reviewers who chose to remain anonymous.

---

## Round 0.3 · accepted · Accept

Thanks for attending the corrections in the corrected version. Therefore, the manuscript is now accepted for publication at PeerJ.